# Putative Complementary Compounds to Counteract Insulin-Resistance in PCOS Patients

**DOI:** 10.3390/biomedicines10081924

**Published:** 2022-08-09

**Authors:** Tabatha Petrillo, Elisa Semprini, Veronica Tomatis, Melania Arnesano, Fedora Ambrosetti, Christian Battipaglia, Alessandra Sponzilli, Francesco Ricciardiello, Andrea R. Genazzani, Alessandro D. Genazzani

**Affiliations:** 1Gynecological Endocrinology Center, Department of Obstetrics and Gynecology, University of Modena and Reggio Emilia, 41121 Modena, Italy; 2Department of Obstetrics and Gynecology, University of Pisa, 56126 Pisa, Italy

**Keywords:** PCOS, insulin resistance, inositols, carnitines, lipoic acid, antioxidants, L-arginine, N-acetyl cysteine

## Abstract

Polycystic ovary syndrome (PCOS) is the most frequent endocrine-metabolic disorder among women at reproductive age. The diagnosis is based on the presence of at least two out of three criteria of the Rotterdam criteria (2003). In the last decades, the dysmetabolic aspect of insulin resistance and compensatory hyperinsulinemia have been taken into account as the additional key features in the etiopathology of PCOS, and they have been widely studied. Since PCOS is a complex and multifactorial syndrome with different clinical manifestations, it is difficult to find the gold standard treatment. Therefore, a great variety of integrative treatments have been reported to counteract insulin resistance. PCOS patients need a tailored therapeutic strategy, according to the patient’s BMI, the presence or absence of familiar predisposition to diabetes, and the patient’s desire to achieve pregnancy or not. The present review analyzes and discloses the main clinical insight of such complementary substances.

## 1. Introduction

Polycystic ovary syndrome (PCOS) is the most common female endocrine-metabolic disorder, affecting 5–21% of women at reproductive age [1].

The diagnosis of PCOS is based on the Rotterdam criteria (2003), which requires at least two out of three criteria: chronic anovulation disorders (oligo- or amenorrhea), clinical or biochemical signs of hyperandrogenism, and the morphology of the polycystic ovary (PCO) at ultrasound (the presence of a follicle number per ovary of ≥20 and/or an ovarian volume ≥10 mL on either ovary using a ultrasound transducers with a frequency bandwidth that includes 8 MHz) [2]. PCO is different from PCOS because it refers only to the morphological aspect of ovaries, and it can be present in other endocrine disorders. A formal diagnosis of PCOS requires that the conditions of thyroid dysfunction, hyperprolactinemia, acromegaly, non-classical congenital adrenal hyperplasia, Cushing syndrome, and ovarian/adrenal androgen-secreting neoplasms have been excluded.

Although there is no mention in these criteria, in the last decades the dysmetabolic aspect of insulin resistance (IR) has been introduced and taken into account as a key feature in the etiopathology of polycystic ovary syndrome. IR is defined as the pathological state in which certain tissues have subnormal biological response to a given concentration of insulin [3], and that leads to compensatory hyperinsulinemia in order to maintain glucose blood concentrations in range. 

IR and consequent compensatory hyperinsulinemia contribute to the endocrine dysregulation, leading not only to the risk of cardiometabolic diseases, like impaired glucose tolerance, type 2 diabetes, dyslipidemia, hypertension, obesity, and metabolic syndrome, but also to the typical clinical signs of hyperandrogenemia, such as acne, alopecia, and hirsutism [4,5,6].

The etiology of PCOS is complex and multifactorial, including genetic and environmental factors, witnessed by the presence of familial aggregation of the disorder [7]. IR, obesity, and familial predisposition to diabetes are the most influent among hereditary factors. Indeed the presence of familial diabetes predispose to a defect of the post-receptor signaling not only for insulin, but also for Follicle Stimulating Hormone (FSH) and Thyroid Stimulating Hormone (TSH) [8,9].

Instead, the environmental factors include prenatal exposure to a hyperandrogenic environment, reduced fetal growth (IUGR), or small for gestational age (SGA), as well as exposure to gestational diabetes during pregnancy and inadequate lifestyle [10].

## 2. Endocrine Profile of PCOS

PCOS is characterized by higher concentrations of Luteinizing Hormone (LH), normal or relatively low FSH levels with increased LH/FSH ratio (>2.5), and higher frequency of LH pulsatile release from pituitary gland. Elevated levels of LH induce an excessive stimulation of ovarian theca cells and consequently an overproduction of ovarian androgens that causes an impaired follicular development [11]. Hyperandrogenism is such an important feature of the syndrome that in the past it was considered necessary for the diagnosis of PCOS (NIH criteria, 1990 as well as AE-PCOS) [12], but in reality, it is not present in all PCOS patients. The excess of androgens arises mainly from ovarian secretion, of which Androstenedione is the principal product, and in part from adrenal secretion, represented by Dehydroepiandrosterone sulfate (DHEAS). Testosterone derives from peripheral conversion from Androstenedione and from ovarian and adrenal production. Cytochrome p450c17*α* is the major androgen-forming enzyme responsible for both adrenal and ovarian production, and in PCOS it is typically overexpressed under the modulation of insulin [8]. Indeed, insulin amplifies the effect of LH within ovarian theca cells, resulting in an over activation of p450c17*α*. In addition, the androgens’ synthesis in theca cells seems to be activated by insulin via multiple pathways, such as phosphoinositide-3kinase (PI3K) signaling or the mitogen-activated protein kinase (MAPK) pathway [13].

Moreover hyperinsulinemia affects the production of insulin-like growth factor-1 binding protein (IGF-1BP) in the liver, leading to an increase of IGF-1 levels, which in turns stimulate ovarian androgen synthesis [14].

Peripherally, testosterone is converted into the more active androgen, Dihydrotestosterone (DHT), by the enzyme 5αreductase, whose excessive activity is at the basis of hirsutism [15].

Aromatase activity in PCOS women is impaired: the ovarian aromatization of androgens in estrogens in granulosa cells is reduced, likely due to the effect of compensatory hyperinsulinemia induced by peripheral insulin resistance [11].

Since the majority of androgens that circulate in blood are bound to Sex Hormone Binding Globulin (SHBG), any condition that decreases the levels of circulating SHBG leads to an excess of free circulating androgens, driving to the onset of clinical manifestations of hyperandrogenism like acne, alopecia, and hirsutism. Liver synthesis of SHBG is reduced not only by hyperinsulinemia, but also by hyperandrogenism [16].

While Estradiol levels are low or normal, plasma levels of Estrone, a weak estrogen, are increased because of peripheral conversion of Androstenedione by aromatase activity, mainly at the adipose tissue level, with the reversal of estrone/estradiol ratio. This hyperestrogenic state might predispose to endometrial proliferation and increased risk for endometrial cancer [17].

Other endocrine features found in PCOS women are hyperleptinemia, reduced adiponectin, decreased opioidergic tone, and excessive activity of kisspeptin-secreting neurons, thus inducing an increase of Gonadotropin Releasing Hormone (GnRH) and LH release, which is pathognomonic of the syndrome [18,19,20]. It has been demonstrated a positive correlation between concentrations of Leptin and clinical-hormonal index of insulin resistance. In addition, Leptin is linked to Neuropeptide Y modulation on the reproductive axis; thus, it has been involved in reproductive disturbance [21,22].

PCOS is the most common cause of anovulation and therefore of infertility. The higher levels of LH, not balanced by FSH, whose activity is amplified by hyperinsulinemia, and the excess of ovarian androgens levels lead to increasing levels of AMPc in granulosa cells and to a disrupted follicle growth. Androgens inhibit follicular maturation with the consequent accumulation of follicles in different growth phases or become atrophic, thus not achieving the final maturation [23] (Figure 1).

Anti-Mullerian Hormone (AMH) serum levels are closely correlated with the number of early antral follicles. AMH is mostly produced by granulosa cells of follicles from 2 to 9 mm in diameter. AMH levels are significantly higher in PCOS patients with hyperandrogenism than without it. This may reflect the severity of disruption of folliculogenesis in patients with hyperandrogenism. Moreover, serum AMH levels have been observed to be higher in women with insulin-resistant PCOS that in patients with normal insulin sensitivity [24,25]. In addition, AMH induces a lower aromatase expression/function, thus improving the hyperandrogenic state.

## 3. Metabolic Profile in PCOS

One of the most frequent features of PCOS patients is the presence of increased insulin plasma levels as a compensation of insulin resistance, present in approximately 70–80% of women with PCOS and central obesity, as well as in 30–40% of lean women diagnosed with PCOS [26,27]. Overweight or obesity status might be present in 50–70% of women with PCOS [21,28] (Figure 1)

The excess of androgens negatively modulates the function of insulin in the liver and at peripheral levels. It has been shown that testosterone affects the transmission of post binding insulin signal, reducing both the number and the efficiency of glucose transporters (GLUT-4), contributing to insulin resistance [29]. Additionally, obesity is associated with a decreased expression of GLUT-4 [30].

The adipose tissue located at the abdominal level, typical of obese PCOS women, compared to gynoid adipose tissue, is metabolically more active, more sensitive to catecholamines and less to insulin, and releases higher amount of free fatty acids. Thus, it can be considered as an index of cardiovascular risk. In fact, a Waist–Hip Ratio (WHR) greater than 0.80 is a marker of android obesity. Moreover, the waist circumference, more than the BMI, is directly correlated with the risk of developing Metabolic Syndrome (MS), with an increased risk in case of waist circumference equal or greater than 102 cm in males and 80 cm in women [31].

PCOS patients might develop impaired glucose tolerance and type 2 diabetes, and the mean age of this diagnosis in PCOS patients is lower than in controls. For this reason, it is suggested to perform a 2 h oral glucose tolerance test (OGTT) in every patient diagnosed with PCOS, especially in those with BMI > 25 [32].

Impaired glucose tolerance is characterized by asymptomatic moderate increase of fasting glucose levels (110–125 mg/dl), which may precede diabetes. Conversion of impaired glucose tolerance to frank diabetes in women with PCOS is 5–10 times more frequent compared with non-PCOS women. Additionally, a family history of diabetes and the presence of obesity are important predictors for the development of type 2 diabetes [33].

Insulin resistance can be assessed with different methods, such as an insulin maximal blood level above 50 µU/mL within 90 min after 75 g of glucose load (OGTT), insulin basal plasma level above 12 µU/mL, or a glucose-to-insulin ratio <4.5 [34]. The principal index of insulin resistance is the HOMA index, which can be computed as homeostasis model assessment of insulin resistance (HOMA-IR) as (fasting insulin mU/l) × (fasting glucose mmol/l)/22.5. A value above 2.5 in adult women is suggestive of insulin resistance [35]. Another evaluation of insulin resistance is estimated as the insulin area under the curve (AUC) of 7000 µIU/mL or more in 120 min [36].

Alterations of lipid profile have been reported in PCOS women with significant increase of low-density lipoprotein cholesterol (LDL-c), total cholesterol, triglycerides, and free fatty acids, and a decrease of high-density lipoprotein cholesterol (HDL-c) [37]. Therefore, it is suggested to evaluate fasting lipid and lipoprotein levels in every PCOS patient. It has also been shown that PCOS women have an increased risk to develop hypertension [38]. Insulin resistance and hyperinsulinemia negatively interact with vascular factors like endothelin and nitric oxide, leading to alterations of vasodilatation and predisposing to hypertension [39].

All these disorders can be connected in the clinical condition of metabolic syndrome, which includes a cluster of metabolic abnormalities such as elevated blood pressure levels (greater than or equal to 130/85 mmHg), increased waist circumference (greater than or equal to 88 cm), elevated fasting glucose levels (greater than or equal to 100 mg/dL), reduced high-density lipoprotein cholesterol levels (less than or equal to 50 mg/dL), and elevated triglyceride levels (greater than or equal to 150 mg/dL) [31]. This syndrome is related to insulin resistance and hyperandrogenism; indeed, the prevalence of metabolic syndrome is higher in hyperandrogenic subjects than non-hyperandrogenic anovulatory women affected by PCOS [40,41].

Another important aspect recently investigated is the increased risk of Non-alcoholic Fatty Liver Disease (NAFLD) in PCOS women. It consists in the hepatic fat accumulation in individuals who do not drink excessive amounts of alcohol, as a consequence of obesity and insulin resistance [42]. The hepatic insulin extraction (HIE) index reflects the balance of the synthesis and clearance of insulin and C-peptide molecules, which derive from the cleavage of the proinsulin released by β-pancreatic cells. While C-peptide has a negligible hepatic extraction, it reflects the pancreatic production, and insulin is mainly cleared by the liver before entering the systemic circulation. The HIE index is computed as the ratio between the area under the curve (AUC) of insulin and the AUC of C-peptide (AUC Ins/AUC C-Pept). Whereas the ideal ratio in the pancreatic vein might be 1, the plasmatic ratio depends on clearance kinetics of the two peptides.

Genazzani et al. recently reported that PCOS patients with familial diabetes have non-optimal liver function, as indicated by higher AST and ALT levels than PCOS patients without familial diabetes. This aspect is frequently associated with high baseline insulin levels and an insulin response to OGTT higher than 50 µU/mL within 90 min. These features are recognized as predisposing factors not only to metabolic syndrome, but also to NAFLD and liver steatosis. In these patients, there is not any HIE decrease at 30–60 min after OGTT, because insulin clearance by the liver is reduced, leaving a higher amount of insulin uncleared in the blood. These data suggest that the compensatory hyperinsulinemia in overweight/obese PCOS is due not only to reduced peripheral insulin sensitivity, but also to reduced hepatic insulin clearance for an impaired expression/synthesis of the insulin degrading enzyme (IDE) [43,44,45].

## 4. Pharmacological Therapies

Obesity and insulin resistance (IR) negatively impact the fertility of PCOS women, since together, these factors decrease the number of spontaneous ovulatory cycles and raise the percentage of spontaneous abortions. For this reason, the first line of treatment for PCOS patients is a correct lifestyle through hypocaloric diet and physical exercise (20–60 min of physical activity a day, from 3 to 5 times a week), in particular in women with BMI > 25 kg/m^2^ [17].

In terms of weight loss, there is no clear evidence regarding the most effective composition of the diet; indeed, it seems that caloric restriction itself, rather than the composition of the diet, is the key factor of the success [7].

Modest weight loss (5–10% of initial body weight) has been shown to lead a rise in SHBG, a reduction of circulating androgens and clinical manifestations of hyperandrogenism, an improvement in ovarian function, and a higher pregnancy rate [33,46].

If the patient is not attempting to conceive, the most effective therapy to treat hyperandrogenism is the administration of combined estro-progestinic pills. Hormonal contraceptives suppress ovary androgen production, blocking the ovarian cycle and increasing SHBG synthesis [47,48]. While the estrogenic compound has only an ovariostatic activity, the progestational compound might have an antiandrogenic action, in particular with Cyproterone Acetate, Dienogest, Drospirenone, and Chrlormadinone acetate [48]. The effect on the signs of hyperandrogenism depends on the skin cell renewal cycle and becomes evident after at least 4 months. For this reason, the minimal duration of the treatment is 4–5 months or more.

The association of antiandrogenic compounds such as Flutamide, Finasteride, or Spironolattone, in combination with oral contraceptives, leads to better results on clinical manifestations of hyperandrogenism [49,50]. However, estroprogestinic pills and antiandrogenic compounds do not act on the metabolic impairments of PCOS patients. When lifestyle measures prove to be unsuccessful, the use of insulin-sensitizing agents seems to be indicated.

Metformin is an oral insulin-sensitizing agent, considered the first line treatment in type 2 diabetes mellitus. It has been widely used to treat insulin resistance in PCOS patients. It is a synthetic biguanide that reduces hepatic glucose production and intestinal glucose uptake and increases glucose uptake at peripheral levels, in particular in skeletal muscle and the liver.

Therapy should be initiated with a starting dose of 250 mg twice a day (15 min before lunch and dinner), which can be increased up to 500–1000 mg after 15 days with a gradual increase in dosage to avoid common gastrointestinal side effects [17].

Metformin has been shown to decrease ovarian and adrenal cytochrome p450c17α, ameliorate hyperandrogenism, decrease the concentrations of androgenic metabolites (androstenedione, testosterone, DHEAS), and restore ovulatory function [51,52,53,54]. Metformin has been proven to be effective on insulin resistance parameters, such as fasting insulin levels, dyslipidemia, Body Mass Index (BMI), oxidative stress, and inflammatory markers [6,55]. Metformin increases the ovulatory cycle and pregnancy rate compared to placebo, reducing peripheric insulin resistance and consequently hyperinsulinemia, as demonstrated by a recent Cochrane review that includes 42 trials [56].

Although Metformin ameliorates ovulation rate, it should not be considered the first line treatment of chronic anovulation since there are specific ovulation inducers, such as Clomiphene Citrate and Letrozole, which have better results in terms of ovulation rate, number of pregnancies, and live births. On the other hand, the addition of Metformin to Clomiphene Citrate should be considered for women with PCOS and significant insulin resistance and obesity [56].

The cost effectiveness of this pharmacological treatment with common side effects, such as nausea, vomiting, abdominal pain, and diarrhea, may reduce subject compliance and limit its use, especially when high doses are needed.

In anovulatory women with PCOS, attempting to conceive, the treatment of choice is Clomphene Citrate (CC), a selective estrogen modulator. The starting dose is 50 mg a day for five days (from the third to fourth to the seventh to eighth day of the menstrual cycle) and it can be raised up to 150 mg a day. The maximum suggested period of administration is six months [57]. Unfortunately, a resistance to Clomiphene occurs in 15–40% of women with PCOS.

Letrozole, an aromatase inhibitor, is used as an alternative to Clomiphene, especially in Clomiphene-resistant women. It is even more effective than Clomiphene but it is burdened by important side effects, and in Italy it is used off-label [17].

When the former treatments fail, the second line treatment for CC-resistant PCOS patients is the surgical laparoscopic ovarian drilling. This technique consists in a partial destruction of ovarian cortex and the consequent drop of the production of androgens, that leads to an increase of FHS and a decrease of LH, and the subsequent improvement of ovary function [58].

Another second line treatment for CC-resistant PCOS patients is the ovulation induction with exogenous gonadotropins. These injective treatments stimulate follicular growth and are usually used in association with in vitro fertilization (IVF) techniques as a third line treatment. Women with PCOS are particularly at risk for ovarian hyperstimulation syndrome (OHSS). Therefore, in order to avoid OHSS and multiple gestation, a close ultrasound monitoring to detect follicle growth is required. With the low-dose protocol, the ovulation and monofollicular development rate is nearly 70%, while the pregnancy rate is 20% per cycle [57].

## 5. Complementary/Integrative Treatments

Recently, different therapeutic approaches have been developed using integrative compounds in order to treat PCOS women, especially those for whom a pharmacologic treatment is not yet advisable, but in whom lifestyle modifications were unsuccessful. The majority of these treatments act at metabolic levels, reducing insulin resistance, which is at the basis of the PCOS. We will discuss these different integrative treatments, which up to date have shown better results in term of metabolic and reproductive function in PCOS women.

### 5.1. Inositols

Inositols are a large family of nine stereoisomers, structurally similar to glucose, belonging to the family of pseudo-vitamin B complex. Inositols are generally found in many plants, legumes, cereals, nuts, and fruits, but they can also be synthetized in the human body [21].

Among the nine isomers, Myo-inositol (MYO) and D-chiro-inositol (DCI) play the most relevant metabolic role in our biology. Myo-inositol is the prevalent form in human tissues, with a plasma MYO/DCI ratio of approximately 40:1, but every tissue has a specific MYO/DCI ratio [59].

MYO endogenously can be synthesized from Glucose-6-phosphate, which is isomerized and then dephosphorylated. D-chiro-inositol is synthetized through the activity of an enzyme, Epimerase, that converts MYO into DCI. Epimerase is stimulated by insulin, and each tissue has a typical conversion rate.

Once MYO enters the cell, it is converted into phosphatidyl-Myo-inositol, a precursor of Inositol-3-phophate, which acts as intracellular second messenger for insulin and TSH and FSH signaling pathways [60,61,62]. This aspect is important since inositol administration can improve both the metabolic and the endocrine function in PCOS patients, taking part in the transduction of the signal of insulin, but also of FSH.

In insulin post-receptor signaling, inositols are implicated in two different pathways. In the first, insulin binds its receptor and recruits insulin receptor substrates (IRS), activating phosphatidylinositol-3-kinase (PI3K), which generates phosphatidylinositol-(3,4,5)-trisphosphate (PIP3). This molecule activates the enzyme PDK, which turns on the protein kinase PKB-Akt. This pathway leads to the translocation of GLUT4 vesicles to the plasma membranes in order to increase glucose transport into the cell, mainly in skeletal and cardiac muscle and adipose tissue [63]. The second pathway is mediated by G-protein (Gp) and through the hydrolysis of glycosylphosphatidylinositol (GPI), it is released an inositol phosphoglycan containing D-chiro-inositol (INS2). INS2 acts at cytosol and mitochondrial levels, but it is also released out of the membrane to amplify its action. Inside the cytosol INS2 stimulates glycogen synthase (GS) directly and indirectly via PI3K/PDK/Akt/GSK3 pathway (Figure 2). The result of this via is the glycogen storage. Therefore, DCI reduces the amount of cytosolic glucose creating a glucose gradient that enhances the uptake of glucose through the mobilization of GLUT4 transporters. In the mitochondria, INS2 activates pyruvate dehydrogenase phosphatase (PDHP) and pyruvate dehydrogenase (PDH), inducing glucose oxidative use, thus amplifying the glucose gradient inside the cytosol.

In this complicated transmission of insulin post-receptor signaling, it is involved another important molecule, alpha lipoic acid (ALA), which we will discuss later.

Considering the relevant role of inositols in promoting glucose uptake, glucose oxidative use, and glycogen storage, their concentrations and functions are essential for the maintenance of the glycemic homeostasis, acting like insulin sensitizers [17].

Interestingly, it has been demonstrated that urinary excretion of DCI is reduced, while MYO urinary content is increased in humans and experimental animals affected by type 2 diabetes [59,64]. Such imbalance in MYO conversion to DCI, expressed as MYO to DCI ratio, was higher not only in type 1 or type 2 diabetic patients, but also in non-diabetic relatives of diabetic patients. A lower concentration of plasmatic DCI with normal levels of MYO was also found in PCOS patients [65]. Thereby the epimerase function was suspected to be impaired in diabetic women and PCOS patients with diabetic relatives, with a consequent decrease of MYO to DCI conversion in insulin-sensitive tissues such as kidney, liver, and muscles [66]. These findings support the hypothesis that diabetes and familial predisposition to diabetes induce an abnormal function/expression of epimerase activity, thus contributing to the insulin resistance and compensatory hyperinsulinemia [67].

Taking into account the importance of DCI and MYO in the transduction of insulin and FSH signal, over the past few years many studies have been done to evaluate inositols as integrative treatment in PCOS patients. Genazzani et al. showed that the administration of 500 mg of DCI to obese PCOS patients improved both metabolic and endocrine functions since LH, LH/FSH ratio, Estradiol, 17-OH-progesteron, and androstenedione were significantly reduced and insulin, the glucose/insulin ratio, and BMI were decreased. When OGTT was evaluated, the hyperinsulinemic response to OGTT improved after the treatment as well as the glucose/insulin ratio, the AUC of insulin, and the maximal insulin response to glucose load. Subdividing patients according to the presence or absence of diabetic first grade relatives, it was found that before treatment, obese PCOS patients with familiar predisposition to diabetes had a greater hyperinsulinemic response to glucose load than those of the other group. DCI administration in PCOS patient with familiar predisposition to diabetes leads to a greater improvement of the insulin response to OGTT. These data support the hypothesis that predisposition to diabetes probably affects epimerase function, leading to a reduction of MYO conversion to DCI, and that DCI administration overcomes this impairment [68].

DCI supplementation has been demonstrated to decrease AMH plasma levels, indicating the reduction of increased functional ovarian reserve typical of PCOS patients [69].

Many studies have evaluated the role of MYO administration on the reproductive function of PCOS patients. Chiu et al. demonstrated that in the ovary MYO concentrations in follicular fluid (FF) has a positive correlation to oocyte quality. Indeed, follicles with higher levels of MYO presented good quality oocyte and higher estradiol concentrations [70]. The same group reported that adding MYO to the culture medium of mouse oocytes improved meiotic progression on oocytes [71].

In another study the administration of MYO (2 g per day) in obese PCOS patients showed endocrine improvements with a decrease of LH, LH/FSH ratio, androstenedione, and BMI. Subdividing the patients according to baseline insulin plasma levels (below or above the cut-off of 12 µU/mL), only patients with hyperinsulinemia (basal insulin level above 12 µU/mL) showed a significant reduction of insulin plasma levels and the area under the curve of insulin [34].

Artini et al. pre-treated PCOS patients undergoing IVF with 2 g of MYO/day and demonstrated an improvement in not only metabolic and endocrine, but also in oocyte quality, recruitment, fertility rate, and delivery rate [72]. An improvement of efficacy in ovulation induction with Clomiphene Citrate, with higher rate of pregnancy and delivery, has been demonstrated by Kamenov et al. with the supplementation of 2 g per day [73]. However, none of these studies considered the presence of familial diabetes in their population.

The administration of MYO to support Assisted Reproduction Technology (ART) in PCOS patients, typically suffering from infertility, has been studied: beneficial effects of MYO have been demonstrated in oocyte maturation, embryo development, and pregnancy rate [74,75,76]. Moreover, the treatment with MYO decreased the rFSH dose required in patients undergoing IVF or ICSI, both in PCOS and in non-PCOS patients, suggesting the use of this integrative compound in ART in order to improve reproductive outcomes and the cost-effective reduction of gonadotropin use [77,78,79].

Unfer et al. reviewed 21 studies on clinical outcomes of MYO as a treatment for PCOS patients. The common results were the improvement in hormonal parameters and in metabolic index like insulinaemia, the HOMA index, BMI, and the glucose/insulin ratio. In the lipid profile, total cholesterol concentrations decreased and high-density lipoprotein concentrations increased. Moreover, menstrual function and fertility improved. The supplementation with MYO increased the bioavailability of the inositol phosphoglycan (IPG) second messenger involved in the insulin transduction, leading to a reduction of insulinaemia and its detrimental role in PCOS syndrome. This review also demonstrated that MYO rather than DCI improved oocyte and embryo quality in FIVET programs, suggesting that ovaries have different metabolic-endocrine pathways than other tissues with a specific MYO to DCI ratio [80].

Differently from other tissues, ovaries can maintain normal insulin sensitivity, despite the presence of insulin resistance. In effect, ovaries never become insulin resistant, and for this reason the hyperinsulinemia enhances ovarian epimerase activity, leading to an excess of conversion from MYO to DCI. This phenomenon is called “ovarian paradox” [81,82]. However, the increase of DCI concentrations seems to enhance androgens synthesis, while MYO depletion worsens FSH signaling and oocyte quality [82,83]. Sacchi et al. in 2016 demonstrated that DCI regulates the gene expression of enzymes involved in steroidogenesis in human granulosa cells, reducing both the expression of aromatase and cytochrome P450 side-chain cleavage (citp450scc) genes and also of IGF-1 receptor synthesys thus countereacting insulin action. Doing so DCI affects estrogen levels without completely blocking their biosynthesis [84]. In contrast, it is assumed that MYO may stimulate aromatase function. Indeed, MYO takes part in FSH post-receptor signaling, which stimulates aromatase synthesis with the conversion of androgens to estrogens and the follicular maturation. As a matter of fact, PCOS is characterized by a relative reduction of FSH and the consequent decrease in aromatase synthesis [85,86].

The ovary paradox may explain why supplementation with high doses of D-chiro-inositol could not have positive effects on ovarian function in PCOS patients. Indeed, the administration of 2400 mg/day of DCI for 6 weeks, on the one hand, improved insulin sensitivity and metabolic parameters, but on the other hand, it leads to a non-significant increase of testosterone [87], thus suggesting that only very high doses of DCI could not affect androgen production. On the contrary, it has been demonstrated that low doses of DCI treatment have a metabolic systemic effect on hyperinsulineamic patients, with a reduction of insulin levels, leading to a reduction of epimerase activity and increased MYO ovarian levels. The increase of MYO improves FSH sensitivity, restoring the ovulation rate [77].

High DCI levels at the ovarian level have been thought to negatively impact the quality of oocytes and blastocyst [81,88], but Sacchi et al. [84] clearly demonstrated the positive relevant role of DCI on various functions at the ovarian level. However, none of these studies considered the potential bias of the presence or absence of familial diabetes in the population studied. Indeed, plasmatic MYO and DCI concentrations and the state of insulin resistance/hyperinsulinemia represent the metabolic aspect of the great part of our biology, mainly metabolically active organs like muscles, liver, and kidney, whereas ovarian environment has a different metabolic setting with its own MYO-DCI concentrations and epimerase activity [21].

Therefore, both MYO and DCI seem to be potentially effective in PCOS patients, with specific different functions according to familiar predisposition to diabetes, and consequent relative function of epimerase, whether the aim of the integration is focused on the restoration of metabolic or reproductive function.

A putative balanced combination dose of MYO and DCI may modulate the hyperinsulinemic metabolism through the action of DCI, and the reproductive ovarian function thanks to the MYO component. Many studies have been conducted on supplementations with different MYO-DCI ratios on animals and PCOS humans. A recent review of MYO and DCI integrative use, through the analysis of the literature available, demonstrated that the 40:1 combination of MYO and DCI produces the most significant improvements, supporting the restore of the physiological plasma concentrations [77,89,90,91]. It is obvious that such combination might be optimal mainly for those PCOS patients without familial diabetes.

Inositols have the same chemical structure as glucose, thus inositols’ intestinal absorption is decreased in the presence of food, and consequently, MYO and DCI should be administered away from meals. Furthermore, also the combination of MYO to DCI generates a competition in the intestinal absorption between the two isomers. Therefore, in a PCOS patient with familial diabetes, the combination of MYO and DCI in 40:1 ratio could lead to an insufficient dose of absorbed DCI.

In conclusion, there are numerous studies on the efficacy of inositols in PCOS treatment, both on metabolic restoration of insulin sensitivity, on endocrine modulation of hyperandrogenemia, and on the improvement of reproductive outcomes. However, it is to underline the importance of choice of the inositol tailored to the patient, according to the anamnestic investigation on diabetes predisposition, metabolic, and endocrine patterns and the primary aim of the treatment.

### 5.2. Alpha Lipoic Acid

Alpha lipoic acid (ALA) is a biological compound provided with potent antioxidant activity, present in vegetables like broccoli, spinach, and potato and mainly in red meat and offal such as heart and liver. ALA is present in two different enantiomeric forms, R-lipoic acid and S-lipoic acid, of which only the naturally occurring R isomers act as essential cofactors in biological systems for mitochondrial enzyme [92].

Inside the mitochondria the redox balance is preserved by an antioxidant defense network, consisting of stress-responsive enzymes such as Superoxide Dismutase (SOD), Catalase, and Reduced Glutathione (GSH). This system is activated in response to excessive production of reactive oxygen species (ROS) in the mitochondria, thereby neutralizing the ROS before they can damage biological molecules. ROS are highly reactive chemical molecules, formed as a natural byproduct of the normal aerobic metabolism of oxygen and when ROS exceed the buffering capacity of the cells, oxidative stress occurs [93].

As lipoamide, ALA is also a cofactor in multienzyme complex that catalyzes the oxidative decarboxylation of alpha-keto acids such pyruvate, alpha-ketoglutarate, and branched chain alpha-keto acids [94].

In cells containing mitochondria, ALA is reduced to dihydrolipoic acid (DHLA), in an NADH-dependent reaction, whereas in cells that lack mitochondria, ALA can instead be reduced to DHLA (dihydrolipoic acid) via NADPH (nicotinamide adenine dinucleotide phosphate) with glutathione (GSH) and thioredoxin reductases [95]. Unlike GSH, which has antioxidant actions only in the reduced form, both the oxidized and reduced forms of ALA are powerful antioxidants whose functions include quenching of reactive oxygen species (ROS), regeneration of exogenous and endogenous antioxidants such as vitamins C and E and GSH, chelation of metal ions, reparation of oxidized proteins, regulation of gene transcription, and inhibition of the activation of NFkB. Moreover, it takes part in the regulation of glucydic and lipid metabolism [17].

Endogenously, lipoic acid is synthesized from octanoic acid by the action of Lipoic Acid Synthase (LASY). Padmalayam et al. demonstrated that LASY expression is downregulated in animal models of type 2 diabetes and obesity, compared with age- and sex-matched controls. This enzymatic defect leads to a reduction of acid lipoic levels in mitochondria, with an alteration of the antioxidant defense system, and to an exacerbation of inflammation. This LASY downregulation also results in decreased glucose cell uptake, increased insulin resistance, and mitochondrial disfunction [93].

ALA has recently been proposed as an adjuvant therapy in diabetes and other endocrinopathies [96,97]. In human and animal models, it has been demonstrated that ALA increases the glucose uptake in skeletal muscle through the activation of AMP-activated protein kinase (AMPK). AMPK is activated in response to ATP depletion, which causes a concomitant increase in the AMP-to-ATP ratio. Once activated, AMPK phosphorylates downstream substrates, the overall effect of which is to switch off anabolic pathways that consume ATP, such as fatty acid and cholesterol synthesis, and to switch on catabolic pathways that generate ATP, like fatty acid oxidation and glycolysis [98]. The activation of AMPK increases glucose uptake as it induces the translocation of GLUT 4 and GLUT1 to cytosolic membrane of adipocyte and muscle cells in a similar way as insulin, and it increases the expression of GLUT4 genes [99].

The insulin sensitivity is increased also through the reduction of triglyceride storage in skeletal muscles thanks to the inactivation of Acetyl-CoA Carbossilase (ACC) and thus to the reduction of fatty acid synthesis and the rise of the fatty acid oxidation. The accumulation of triglycerides in skeletal muscles contributes to insulin resistance in obesity and type 2 diabetes [100,101].

Several clinical trials have also demonstrated the improvement of insulin sensitivity in insulin-resistant and/or diabetic patients treated with the antioxidants vitamin C, ALA, vitamin E, and glutathione [102]. In patients with type 2 diabetes, both acute and chronic administration of ALA have been demonstrated to reduce insulin resistance [101,103].

Recent studies showed that in PCOS women, oxidative stress is increased for the higher free radical production and the decline of antioxidant agents’ levels and their enzymatic activity. The increased oxidative status seems to worsen insulin resistance [104]. This suggests that a reduction of alpha lipoic acid could determine insulin resistance, and that a supplementation with ALA could in turn be useful in the treatment of metabolic and reproductive disorders in PCOS patients.

On these assumptions, many studies have been conducted to evaluate the effects of ALA administration on insulin resistance and on hormonal parameters in PCOS women.

Genazzani et al. treated obese PCOS women with 400 mg/day of ALA for 12 weeks and observed in all patients a significant improvement of parameters such as: insulin, glucose, GOT, BMI, the HOMA index, insulin response to glucose load (OGTT), insulin maximal response (Δmax), and AUC. Subdividing the patients in two groups according to the presence or absence of diabetic relatives, it was identified that only the group with diabetic relatives has a significant decrease also in GOT and triglycerides. These finding suggest that ALA has specific efficacy in the liver, reducing the risk of developing a liver impairment such as Non-Alcoholic Fatty Liver Disease (NAFLD) [105]. Moreover, the amelioration of insulin sensitivity in PCOS with diabetic relatives under ALA treatment demonstrates that ALA integration might overcome the possible defect of endogenous synthesis of ALA due to the impairment of the enzyme LASY. It should be observed that with ALA administration no changes in hormonal or reproductive parameters were observed, indicating that ALA acts only on the metabolic side.

These data in PCOS obese women are in agreement with the findings of Yi Yang, in one animal model of hyperinsulinemic mice, where the administration of ALA ameliorated metabolic functions [106]. Patients with PCOS, in particular the ones with metabolic syndrome, have higher levels of the hepatic steatosis index, characteristic of NAFLD. The obesity, insulin resistance and type 2 diabetes are not only prominent metabolic features of PCOS, but also the principal risk factors for NAFLD [105].

ALA administration in PCOS patients leads to an improvement of hepatic functional index, suggesting its role in the amelioration of hepatic mitochondrial performance [107]. All these findings together, especially those on PCOS subjects, are clear demonstration of the efficacy of ALA treatment, absolutely optimal to counteract the reduced insulin sensitivity.

Recently it has been studied also the association of ALA with inositols. The administration of ALA (400 mg/day) and MYO-inositol (1 g/day) to overweight/obese PCOS patients proved to enhance insulin sensitivity through a decrease of insulin response to the oral glucose tolerance test and HOMA index, and proved to be a hormonal asset with a reduction of LH and the LH/FSH ratio. Subdividing patients according to hyperinsulinemic or normoinsulinemic response to oral glucose load (<50 µU/mL insulin levels within 90 min after glucose load), only hyperinsulinemic patients showed a significant reduction of insulin release and the HOMA index. Subdividing according to the presence or absence of familial diabetes, only women with diabetic relatives showed significantly reduced basal insulin levels, similar to hyperinsulinemic PCOS [108].

In addition, in the study of De Cicco, the administration of higher dose of ALA (800 mg/day) and MYO-inositol (2 g/day) for 6 months leads to a reduction of clinical and biochemical hyperandrogenism, BMI, restoration of ovulation rates, and a rise in SHBG [109].

The association of D-Chiro inositol with ALA was also evaluated. The administration of DCI (500 mg/day) and ALA (300 mg/day) for 12 weeks in overweight/obese PCOS patients showed a significantly change in LH, androstenedione, insulin, LDL plasma levels, BMI, the HOMA index, maximal insulin, and C-peptide response after OGTT. Regarding the presence or absence of familial diabetes, only the group with familial diabetes had a significant reduction of triglycerides, total cholesterol, LDL, GOT, and GPT concentrations. These data are in line with other findings on the important role of ALA in preventing liver damages and NAFLD [110].

Furthermore, Cianci et al. obtained an improvement of HOMA-IR, insulin levels, lipid profile, BMI, and frequency of menstrual cycles with the association of DCI (1 g/day) and ALA (600 mg/day) [111].

Considering all these data together, it comes out the optimal effects of ALA when combined with MYO or DCI, according to the typology of PCOS patients. Such data sustain the relevant role of ALA on IR and reveal how the coupling with inositols is optimal. ALA results as a key element as integrative treatment for PCOS, especially when the anamnestic investigation discloses not only the overweight/obesity, but also first grade diabetic relatives, or familial predisposition to diabetes.

In fact, in these PCOS subjects, the concomitant reduced expression/function of epimerase and LASY can be resolved only by the combination of ALA plus DCI as an integrative treatment, as previously reported [17,21,24,108,109,110,111,112].

All these data sustain the relevance of the use of ALA alone for PCOS to counteract IR. Though Laganà et al. [113] did not recognize efficacy in the use of ALA, its combination with inositols (MYO or DCI) represents an extremely efficient treatment. Clinical data [17,21,24,108,109,110,111,112] support the use of combinations as putative strategy to avoid the use of Metformin or to reduce the doses of Metformin if too high, thus reducing the side effects. In fact, while ALA efficiently improves insulin sensitivity, its combination with inositols (MYO or DCI) significantly acts on both reproductive functions and on insulin sensitivity. Indeed, MYO is involved not only in the intracellular signaling of insulin, but it also takes part in the post-receptor pathway of FSH.

Obviously, the choice of the most appropriate inositol is fundamental and based on a well-conducted anamnestic investigation [21,112,114].

### 5.3. Carnitines

Carnitines are quaternary amines introduced by food or synthetized in the body [115]. They are vitamin-like substances, with two enantiomeric forms: L-carnitine, the leading active form involved in cellular energy production; and D-carnitine, an inactive toxic molecule. Furthermore, there are different carnitine esters such as Acetyl-L-carnitine (ALC) or Propionyl L-carnitine [116]. L-carnitine is synthetized from the essential amino acids Lysine and Methionine, largely in the liver, kidney, and brain [117].

Red meat, like beef or lamb, is the main source of L-carnitine, but lower concentrations are present also in fish, pork, poultry, and dairy products. Considering plant-origin products, only avocado and asparagus contain higher concentrations of carnitines [118].

L-carnitine is a cofactor of enzymes, such as Carnitine Palmitoyl Transferase 1 (CPT1), Carnitine Acyltranslocase, and Carnitine Palmitoyl transferase2 (CPT2), involved in the transport of fatty acids in the mitochondria, which leads to the production of a molecule of Fatty Acyl-CoA that undergoes the beta oxidation, producing Acetyl-CoA. Acetyl-CoA also derives from the oxidation of carbohydrates (glucose and lactate) through the action of pyruvate dehydrogenase complex (PDH). Acetyl-CoA finally enters the tricarboxylic acid (TCA) cycle and generates energy as ATP. The PDH complex is the key rate-limiting step in carbohydrate oxidation, according to concentrations of substrates (pyruvate and CoA) and products (Acetyl-CoA). PDH is inhibited by the increase of Acetyl-CoA/CoA ratio with a reduction of the glucose oxidation rate. The Acetyl-CoA/CoA ratio depends on the rate of removal of intramitochondrial Acetyl-CoA by the TCA cycle and the rate of fatty acid beta-oxidation [119,120]. L-carnitine can also transport acetyl-CoA from the mitochondrial matrix to the cytosol.

To sum up, carnitine plays an important role in both carbohydrate and lipid metabolism, as it regulates the intramitochondrial Acetyl-CoA/CoA ratio by modulating the store of Acetyl-CoA from both the PDH complex and beta-oxidation of fatty acids.

Furthermore, carnitine supplementation proved to decrease pro-inflammatory cytokines, such as interferon-γ, tumor necrosis factor-α (TNF-α), interleukins-2 (IL-2), and IL-6, with an anti-inflammatory effect. It also prevents DNA damage caused by free radicals [121]. In addition, L-carnitine takes part in the maintaining of the cell membrane stability through its role in the acetylation of membrane phospholipids and its amphiphilic action. Moreover, carnitines promotes cellular proliferation and decreases apoptosis through their stimulating action on mitochondria [122].

The main gynecological application of carnitines is the treatment of functional hypothalamic amenorrhea (FHA), where they can blunt the negative effect of stress-induced beta-endorphin release, acting on the protein/hormonal functions in the Opioidergic pathway, on Neuropeptide Y and on Pro-opiomelanocortin (POMC) [122]. Carnitines can also interfere with ROS overproduction caused by excessive diet and physical exercise [122].

PCOS is known to be associated with an increased oxidative stress, with a reduction of total antioxidant levels and an increase of free radicals and reactive oxygen species (ROS). A significant reduction in oxidative stress was observed in type 2 diabetic women after the supplementation of 2 g/day of L-carnitine for 3 months [123]. We should point out that the combination of carnitines with antioxidants such as N-acetyl cysteine (NAC) and L-arginine has been reported to greatly improve insulin sensitivity in PCOS patients all along 6 months of treatments [124]. In fact, such a combination has been reported to improve insulin sensitivity acting both peripherally and on liver function since the hepatic insulin extraction (HIE) decreased significantly. Such evidence suggests that a specific role of antioxidants has been played on the expression/function of the insulin degrading enzyme (IDE). The integrative administration of carnitines with NAC and L-Arg restored the liver ability to degrade insulin as demonstrated by the significantly decreased insulinemia with minimal changes in C-peptide plasma levels [124].

Frenkci’s et al. showed that non-obese PCOS women have lower total serum L-carnitines levels, beyond higher androgen levels, compared to healthy women. These data suggest that the reduced circulating and tissues carnitine levels, probably due to the impairment in mitochondrial function, might be involved in the pathogenesis of insulin resistance [125]. In line with these findings, Molfino et al. treated patients with impaired fasting glucose or type 2 diabetes with L-carnitine (2 g twice a day) and hypocaloric diet, obtaining an improvement in insulin sensitivity (HOMA-IR and plasma insulin levels) [126].

Since insulin resistance seems to be correlated to mitochondrial disfunction with a possible reduction of fatty acid oxidation, L-carnitine supplementation has been supposed to improve insulin sensitivity in PCOS patients thanks to its effect on beta oxidation of fatty acids and carbohydrates metabolism. In fact inefficient oxidative phosphorylation increases the oxidative stress and leads to the accumulation of triglyceride in skeletal muscle, which takes part in the pathogenesis of insulin resistance [127,128]. Consistent with this hypothesis, Samimi et al. found that the administration of L-carnitine reduces body weight, BMI, waist and hip circumference, and glucose in PCOS patients [129].

As previously mentioned, when overweight/obese patients were supplemented with a daily association of ALC (250 mg), L-carnitine (500 mg), L-arginine (500 mg), and N-acetyl cysteine (50 mg) for 6 months, and a significant improvement of both insulin plasma levels and of insulin response to OGTT were reported together with amelioration of total cholesterol, HDL, triglyceride, plasma insulin, and HOMA index. Moreover, subdividing according to the normo- or hyperinsulinemic response to glucose load (hyperinsulinemic response is recognized when above 50 µU/mL), the hyperinsulinemic group, who had a reduced insulin hepatic extraction (HIE) index at baseline condition, showed the highest improvement in insulin sensitivity, and showed a significant reduction of AUC HIE [124]. Such data clearly support the efficacy of the integrative treatment with carnitines combined to antioxidants, such as N-acetyl cysteine and L-arginine, that improved hepatic insulin degradation and consequently peripheric insulin sensitivity.

### 5.4. N-acetylcysteine and L-arginine

N-acetyl-L-cysteine (NAC) is commonly used as a safe mucolytic drug and for acetaminophen toxicity, but at higher doses it increases the cellular levels of reduced glutathione (GSH), and it scavenges free radicals, such as hydrogen peroxide and superoxide, acting like an antioxidant agent. It inhibits apoptosis induced by oxidative radical stress [130]. Moreover, it has been shown to increase insulin sensitivity in vivo [131].

Fulghesu et al. demonstrated that the administration of 1.8 g/day of NAC to PCOS women induced a significant fall in testosterone levels and in free androgen index values. Considering patients according to their insulinemic response to OGTT, normoinsulinemic subjects and placebo-treated patients did not show any modification of the above parameters, whereas a significant improvement in insulin sensitivity with a reduction of circulating insulin levels and secretion after OGTT was observed in hyperinsulinemic subjects, who were compromised from a metabolic point of view [132].

PCOS is associated to insulin resistance and endothelial dysfunction, which can be explained by a reduction of nitric oxide (NO) availability due to a reduced protein kinase PKB-Akt activation [39,133]. Interestingly, there is also evidence from animal studies that NO plays a role in oocyte maturation and ovulation. Furthermore, in patients with type 2 diabetes, another condition characterized by insulin resistance and endothelial dysfunction, reduction in NO availability has been demonstrated, which could reflect the increased free radical production connoting the hyperglycemic condition. Masha et al. demonstrated that the administration of N-Acetyl Cysteine (1200 mg) and L-Arginine (1600 mg) for six months determines an increase in the number of menstrual cycles and an improvement of insulin sensitivity with a decreased HOMA index in patients with PCOS [134]. These data support the hypothesis that NAC and ARG have an effect by increasing NO availability and its ability to improve ovarian function.

In addition, a recent review conducted by Sandhu et al. demonstrated that, when used in combination with clomiphene citrate or letrozole, NAC increases ovulation and pregnancy rate in infertile females suffering from PCOS and positively affects the quality of oocytes and number of follicles ≥18 mm [135].

Nitric oxide (NO) is synthesized by nitric oxide synthase (NOS) during the conversion of L-Arginine to citrulline using oxygen and NADPH as the cofactors. Arginine is the substrate for NOS during NO synthesis; thus, the bioavailability of this amino acid is crucial to NOS action. As a semi-essential amino acid, L-arginine can be synthesized in the human body from other amino acids or derives from foods such as peanuts, walnuts, meats, seafood, and legumes such as soybean and chickpeas [136].

On this basis, Rad et al. suggested that L-arginine could have positive effects on glycemic metabolism thanks to its promoting action on NO synthesis and inducing an increase adiponectin secretion in the adipose tissue. Adiponectin might increase insulin sensitivity by activating the AMP-activated protein kinase (AMPK) signaling pathway, which can improve glucose uptake and utilization by the muscles [137].

As a result, the supplementation with L-arginine might have positive metabolic effects in hyperinsulinemic PCOS patients, and the combination with NAC further improves insulin sensitivity, as discussed previously. In fact, while L-arginine improves NO synthesis, NAC acts positively on the synthesis of nytroso-glutathione, thus reactivating reduced gluthahtione to further eliminate ROS [124].

### 5.5. Melatonin

Melatonin is an indoleamine hormone released by the pineal gland, whose production and secretion are promoted at night in response to darkness, since light can suppress its release. Melatonin has been identified to have different pharmacological properties such as antioxidant, immunomodulatory, anti-angiogenic, and oncostatic effects [138].

Melatonin seems to be relevant for PCOS patients since it is a potent free radical scavenger that exerts protective effects in female reproductive organs. There are data that sustain that melatonin positively acts on follicular maturation and ovulation in PCOS through the protection of follicles against oxidative stress and their rescue form atresia [139].

In PCOS patients, the melatonin level in serum has been found higher than in healthy women while its concentrations in the follicular fluid resulted in reduced PCOS (FF). This fact has been demonstrated to be due to a decreased uptake of melatonin in ovarian follicles in PCOS patients, thus resulting in lower follicular fluid melatonin levels when compared to the healthy condition [140].

On this basis, melatonin administration has been proposed to compensate the reduction of this hormone in follicular fluid and can halt ovulation problems [141]. In fact, Pacchiarotti et al. demonstrated that the oral administration of melatonin does increase the content of melatonin in the FF and has a synergistic effect with Myo-inositol to promote oocyte development and follicular discharge [142].

After six months of melatonin therapy in 40 normal-weight PCOS patients, menstrual irregularities and hyperandrogenism were improved. The lack of significant alterations in the secretion of insulin and insulin sensitivity suggests that melatonin may act on the ovary through an independent mechanism [143,144].

Melatonin treatment in PCOS patients significantly affects body characteristics including reduced body weight, body mass index, and intra-abdominal fat [145].

A recent review shows that melatonin administration plays a positive effect and leads to better assisted reproductive outcomes by modulating the activity of some enzymes, such as antioxidant enzymes and aromatase; regulating lipid metabolism; improving endocrine hormone levels like raising FSH, lowering LH, and androgen levels; and relieving insulin resistance, as well as reducing inflammatory states [146,147,148].

### 5.6. Berberine

Berberine (BBR), an isoquinoline alkaloid, present in nature in different herbal substances with a long tradition in Ayurvedic and Chinese medicinal systems. It is known to have potent antimicrobial activity against bacteria, fungi, protozoans, viruses, helminths, and chlamydia [149].

In recent years, berberine has received interest for its biological activities. Indeed, different studies demonstrated a wide spectrum of berberine pharmacological effects, such as antihypertensive, antiarrhythmic, antihyperglycemic, anticancer, antidepressant, anxiolytic, neuroprotective, antioxidant, anti-inflammatory, analgesic, hypolipidemic activity, and other effects [150]. One of the major disadvantages of berberine is its poor oral bioavailability, which is attributed to its low aqueous solubility and dissolution [151]. For this reason, most of the berberine remains within the gastro-intestinal lumen. Moreover, the great part of berberine is distributed in tissues and the plasmatic concentrations are very low [152].

Recent studies have demonstrated the utility of berberine in PCOS women, even if the mechanism of this alkaloid in the treatment of PCOS is still unclear and more studies are needed.

In PCOS women, berberine can modulate different metabolic aspects and can it decrease insulin resistance, enhancing the phosphorylation of insulin receptor and insulin receptor substrate-1 (IRS1) in adipocytes [153].

Besides, Lee et al. demonstrated, in an animal model, that berberine enhances AMPK activity in adipocytes and myotubes, it reduces lipid accumulation in adipocytes cells, and it leads to the GLUT4 translocation in muscle cells, thus contributing to glucose lowering [154]. The effects of berberine on GLUT 4 expression has been evaluated also by Zhang et al., who demonstrates that BBR in PCOS rat model is associated with an increase in signal PI3K/AKT and an enhancement of GLUT4 expression [155]. In this way, berberine not only reduces insulin resistance, but also reduces blood sugar levels, achieving an important therapeutic effect in hyperinsulinemic PCOS.

BBR is also effective in the reduction of total serum androgens through the increase of SHBG levels. It has been demonstrated by An et al. that the administration of berberine for three months to PCOS women who are refractory to standard ovulation inductors or who have co-existing infertility factors, led to a significant reduction in total testosterone levels and free androgen index and an increase in SHBG levels [156]. Additionally, Wei et al. obtained an important increase in SHBG with the administration of BBR in association with cyproterone acetate in PCOS women [157].

In addition, Li et al. demonstrated that the administration of 0.4 g of berberine three times per day for four months to anovulatory Chinese women with PCOS leads to a decrease in the SHBG levels in the whole group, and subdividing patients according to the BMI, the reduction was in the normal weight group only [158].

PCOS women, as previously discussed, are characterized by metabolic dysregulation and dyslipidemia. BBR seems to have positive effects by reducing synthesis of triglycerides and ameliorating sensitivity to insulin [159]. Moreover, BBR acts by decreasing low-density lipoprotein cholesterol, total cholesterol, and triglycerides as statins, and by increasing the liver expression of low-density lipoprotein receptor, LDLR [160].

In conclusion, berberine has been shown to have several positive actions if used in the integrative treatment of PCOS and is burdened with little side effects, such as diarrhea, constipation, abdominal pain occurring, if overdosed [157]. It is important to remember that BBR crosses the placenta and it is transferred into breast milk, for these reasons it is recommended caution in its administration in pregnancy and breastfeeding [150]. More studies are needed to understand the correct and effective dosage of berberine should be administered in PCOS women.

## 6. New Perspectives in Integrative Treatments for PCOS Patients

### 6.1. Tocotrienols

Vitamin E is a group of eight compounds: α-, β-, γ-, and δ-tocopherol (TF) and α-, β-, γ-, and δ-tocotrienols (T3), which are lipid-soluble compounds [161]. Tocotrienols (T3) differ from tocopherols by the presence of three unsaturated bonds at the positions of 3, 7, and 11 of the side chains (Figure 3). Tocotrienols can be found in palm and rice bran oil, wheat germ, barley, oats, hazelnuts, maize, and in annatto oil [162].

Many studies have established the role of T3 in protecting against metabolic, diabetic, and cardiovascular pathologies [163]. T3s (especially γT3 and δT3) were demonstrated to improve glycemic control in in vitro, in animals, and in the human population [164,165,166]. A limited number of animal studies suggested that T3 reduces body weight or body fat [167]. Both Tocotrienols and Tocopherols could scavenge the free radicals directly by donating the phenolic hydrogen of the chromanol ring. T3 has better membrane antioxidant activity as compared to TF [168]. T3 can suppress cholesterol synthesis by inhibiting 3-hydroxy-3-methyl-glutaryl-coenzyme A reductase (HMGCR) post-transcriptionally, the rate-determining enzyme in the mevalonate pathway [169]. It has been shown to downregulate peroxisome proliferator-activated receptor γ (PPARγ), which is the transcription factor critical in adipocytes differentiation [170]. It can prevent the activation of nuclear factor-κB (NF-κB), thereby halting tissue inflammation [171].

Recently, Meganathan reported that the supplementation of T3 in various population groups triggered beneficial effects in cardiovascular health, cancer, immune modulation, neuroprotection, and skin protection. One of the major limitations of Tocotrienols debated in human trials was the lower bioavailability in plasma and the differences in their study designs, populations, formulations, and dosing regimen. Some of these functions have been confirmed in humans, while others are still under investigation [172].

In a recent study, the administration of 300 mg/day of Delta-tocotrienol for 12 weeks in pre-diabetic subjects was found to have a significant effect in improving glycemic control parameters [173]. In addition, Mahjabeen et al. demonstrated that δT3 supplementation in addition to oral hypoglycemic agents in type 2 diabetes patients, improved glycemic control, inflammation, oxidative stress, and miRNA expression without any adverse effect. Thus, δT3 might be considered as an effective dietary supplement to prevent long-term diabetic complications [174].

Moreover, Pervez et al. studied the supplementation with δ-tocotrienol (300 mg twice a day for 24 weeks) in NAFDL patients, and they found out that δ-tocotrienol significantly reduced the fatty liver index, HOMA-IR, high sensitivity-CRP, malondialdehyde (a lipid peroxidation marker), ALT, and AST. δ-tocotrienol effectively improved biochemical markers of hepatocellular injury and steatosis in patients with NAFLD [175].

Up to now there are no studies on the administration of Tocotrienols on PCOS patients, but Tocotrienols have been demonstrated to have anti-hyperlipidemic, anti-hyperglycemic, anti-inflammatory, and antioxidant effects; hence, T3 supplementation might be considered as an integrative therapeutic option in the management of patients with PCOS, especially in case of insulin resistance.

### 6.2. Decaffeinated Green Coffee

Experimental studies illustrate that coffee consumption shows antioxidant properties through the modulation of gene expression of some inflammatory proteins. Additionally, decaffeinated green coffee has not been studied in PCOS patients, but it has shown positive effects, which can help also in dysmetabolic PCOS women. Vitaglione et al. showed that the addition of decaffeinated coffee to a high fat diet (HFD) in rats determined a reduction in hepatic fat accumulation, systemic and liver oxidative stress, and liver inflammation [176]. Moreover, coffee consumption is associated with higher insulin secretion, insulin sensitivity, and β-cells function [177,178].

A randomized controlled trial demonstrated that six months of supplementation with the combination of Berberis aristate (containing berberine), Elaeis guineensis (Tocotrienols), and decaffeinated green coffee by Coffea canephora in patients with Non-Alcoholic Fatty Liver Disease (NAFLD) induces an increase of insulin receptor levels, with the improvement of insulin resistance and hepatic steatosis [179].

Another study was conducted to evaluate the effects of this combination of plant extracts, composed by Berberis Aristata, Elaeis Guineensis, and decaffeinated green coffee from Coffea Canephora, added to a high fat diet (HFD) in a mouse model of NAFLD. This combination of plant extracts has been demonstrated to exert a protective action on obesity, hepatic steatosis, insulin resistance, and dyslipidemia, with a positive effect on miR-122 and miR-34a expression in the liver and on the gut microbiota [180].

Decaffeinated green coffee bean extract (GCE) has been studied in the dose of 400 mg/day in patients with Metabolic Syndrome for 8 weeks. GCE administration had an ameliorating effect on some of the Metabolic Syndrome components such as high systolic blood pressure, high fasting blood sugar levels, insulin resistance, and abdominal obesity [181].

### 6.3. Gymnema Sylvestre

Gymnema Sylvestre is a plant, popularly known as “gurmar”, which grows in India, Africa, and Australia, used in ayurvedic and homeopathic systems of medicine as a potent antidiabetic drug. It is also used in the treatment of asthma, eye complaints, and inflammations. In addition, it possesses antimicrobial, anti-hypercholesterolemic, and hepato-protective activities [182]. The effects of this plant are due to the presence, on its leaves, of active ingredients referred to as gymnemic acids [183].

Bhansali et al. demonstrated that the administration of 50, 100, and 200 mg/kg of deacyl gymnemic acid (DAGA), one of the most active constituents of Gymnema Sylvestre, in rats with metabolic syndrome, reduces plasma glucose and insulin levels with a decrease of HOMA-IR. Moreover, DAGA used in experimental animals, reduces systolic blood pressure, and improves the lipid profile [184]. The reduction of glucose plasma levels seems to be explained by the reduction of glucose absorption in the intestine since the gymnemic acid molecules bind the sodium-dependent glucose transporter 1 in the external layers of the gut, avoiding glucose intestinal absorption in rat models [185].

Whereas Zuniga et al. reported that the administration of Gymnema Sylvestre in patients with metabolic syndrome decreases body weight, BMI, and low-density lipoprotein cholesterol levels, with no modification on insulin sensitivity and insulin secretion [186].

The ability of G. sylvestre to lower blood glucose concentrations has been tested as a hypoglycemic agent in combination with insulin in humans, with encouraging results. A preliminary study shows that administration of 200 mg/day of G. sylvestre extract decreased the required insulin dose by 50% and lowered HbA1c in both type 1 and type 2 diabetes. It also increased the number of beta cells in the pancreas and therefore the internal production of insulin. When 400 mg/day of this extract is taken with conventional hypoglycemic drugs, such as glyburide or tolbutamide, some patients were able to reduce the dose of the drug or even discontinue its use [187].

Gymnema Sylvestre has shown its safety in different studies on its toxicity, but high doses may lead to side effects, such as hypoglycemia, weakness, shakiness, excessive sweating, and muscular dystrophy [188]. The studies currently present in the literature on Gymnema Sylvestre are aimed to investigate its effects on the metabolic syndrome, which is present in most women with PCOS.

## 7. Conclusions

PCOS is a multifactorial syndrome characterized by multiple metabolic and endocrine impairments that need to be investigated through a good anamnesis focused on clinical and family history, and a medical examination to evaluate not only the hormonal impairments (i.e., hyperandrogenic signs), but also the dysmetabolic ones and the eventual presence of hyperinsulinemic signs. It is of great relevance to establish the patient’s BMI, the presence of familial predisposition to diabetes, and the patient’s desire to achieve pregnancy or not, since the correct therapeutic strategy should be patient tailored.

First of all, it is necessary to start with changes in the lifestyle to promote overall health, and in association to this, it can be administered integrative or pharmacological therapies. As a matter of fact, obesity is one of the risk factors for insulin resistance; therefore, weight reduction is one of the first strategies to get therapeutic results, possibly in association to integrative treatment, which also improves insulin resistance reduction.

The presence of familial diabetes is essential to be investigated, in order to choose the most suitable supplementary treatments, since predisposition to diabetes leads to an impaired expression/function of specific components of insulin signaling or enzyme, such as epimerase and LASY. In fact, among the inositols, DCI is the one with a greater efficacy on PCOS women with diabetic relatives, while MYO should be chosen, if there is no predisposition to diabetes. However, both inositols are relevant integrative approach to treat infertility problems. Similarly, carnitines, especially if combined with NAC and L-Arg, demonstrated great efficacy on insulin resistance, also having a positive impact on liver function.

In conclusion, a wide range of compounds are available as complementary substances that can be used to overcome insulin resistance. It is clear that the use of complementary treatments needs to be attentively evaluated according to the clinical conditions of the PCOS patient following an adequate lifestyle.

## Figures and Tables

**Figure 1 biomedicines-10-01924-f001:**
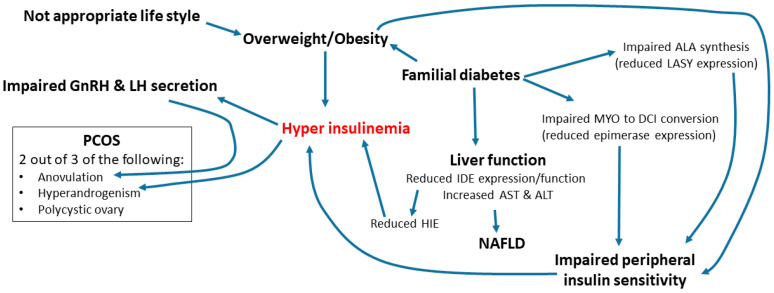
Schematic representation on the role of metabolism impairments in triggering the endocrine impairments of PCOS. An inappropriate lifestyle and/or the coupling with familial diabetes are the main triggers of the insulin resistance that activates the compensatory hyperinsulinemia. Various reduced enzymatic expressions occur in case of familial diabetes and are responsible of the insulin resistance due to peripheral reduced insulin sensitivity.

**Figure 2 biomedicines-10-01924-f002:**
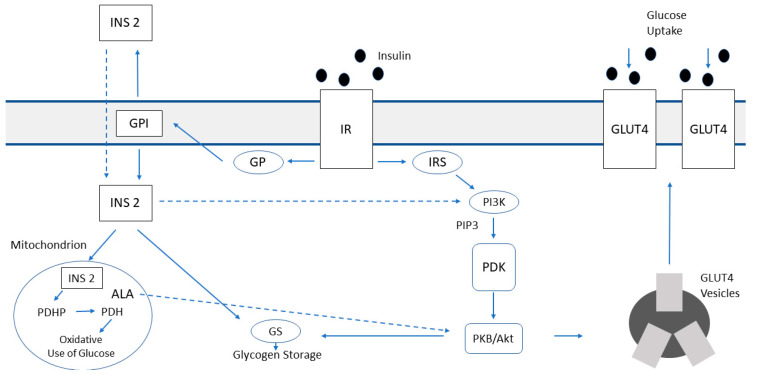
Representation of the role of inositols and alpha lipoic acid in insulin post-receptor signaling, modified from [17].

**Figure 3 biomedicines-10-01924-f003:**
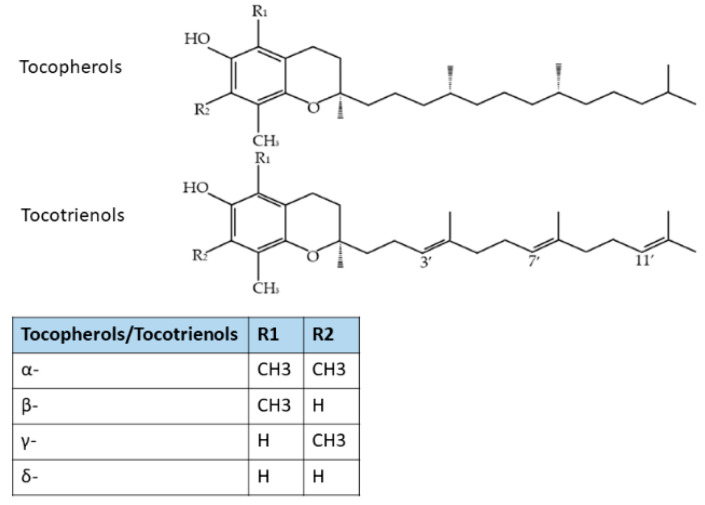
Molecular structures of Tocopherols and Tocotrienols. Modified from [163].

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
