# Peer review of "Putative Complementary Compounds to Counteract Insulin-Resistance in PCOS Patients"

_biomedicines, 2022, doi:10.3390/biomedicines10081924_

Round 1
Reviewer 1 Report
The authors present a literature review entitled “Integrative strategies to counteract insulin-resistance in PCOS patients”. Indeed, the manuscript covers much more metabolic and reproductive aspects of PCOS then insulin-resistance considering integrative therapies. In general, the review fits the scope of the Journal and is of interest to the readers.
The main suggestions are as follows:
The article and abstract do not provide the pronounces objectives of the review and search strategy& information sources (databases with dates of coverage) used when preparing the manuscript. Please consider it to be revised.
1. Introduction.
Please, consider, that the latest guidelines recommended when “Using endovaginal ultrasound transducers with a frequency bandwidth that includes 8 MHz, the threshold for PCOM on either ovary, a follicle number per ovary of ≥20 and/or an ovarian volume ≥10 ml on either ovary, ensuring no corpora lutea, cysts or dominant follicles are present”( doi:10.1093/humrep/dey256). The threshold presented in the manuscript ( lines 31-32) is recommended if using the older US technology.
We suggest to add acromegaly to be excluded as a condition with similar clinical signs with PCOS (lines 34-36).
2. Endocrine profile of PCOS
Please, mention that not only NIH 1990 but also AE-PCOS 2006 agreement considered hyperandrogenism as the main criteria for PCOS diagnosis (lines 62-63).
3. Metabolic profile in PCOS
We suggest mentioning that the latest studies on metabolic syndrome in PCOS patients mainly explore the lower criteria regarding waist circumference for women – 80 cm (line 131-134).
Could you please provide a reference to the phrase “Insulin resistance and hyperinsulinemia negatively interact with vascular factors like 158 Endothelin and Nitric Oxide, leading to alterations of vasodilatation and predisposing to 159 hypertension” (lines 158-160).
Regarding the state that “the prevalence of metabolic syndrome is higher in hyperandrogenic subjects” (lines 167-168), more recent data could be added (DOI: 10.3389/fendo.2022.825528 ; for example)
Sections 4-12.
The main limitations of these sections are as follows: the absence of analysis of study design and levels of evidence of trials, performed in humans, and presentation of data obtained from non-PCOS patients in many cases.
Conclusions.
It is difficult to agree that the integrative therapies “can be considered both alone as well as in association with a pharmacological approach” (lines 910-912), because the term “integrative medicine” presume being combined with conventional medicine.
There are not enough data in the manuscript to support the classification of integrative compounds by the scientific evidence. Therefore, we suggest excluding this classification from the current version.
Author Response
Referee # 1 (changes highlighted in yellow)
This article has not been intended as a meta-analysis or a stringent literature revision but as a clinical methodological upgrade in regards to the the putative use of complementary substances. That’s why no databases sources where cited. The aim was to give a clinical overview on the integrative approach to PCOS mainly finalized to the chance to improve insulin sensitivity as stated in the title.
Introduction
The commenti s correct. We have modified the indications in regards of the ultrasound sensitivity to discriminate for the presence of PCOS. The reference has been added accordingly.
The citation to acromegaly has been added amomng the excluding criteria, as suggested.
Endocrine profile of PCOS
The commenti is correct. Specific citation was added to AE-PCOS criteria.
Metabolic profile of PCOS
As suggested, the waist circumference for women has been corrected
The commenti s correct. A reference has been added to sustain the role of insulin resistance and hyperinsulinemia to trigger alterations of vasodilatation and hypertension
As suggested, additional reference has been added in regards to the prevalence of metabolic syndrome in hyperandrogenic subjects
The comment has been attently considered. We disagree with the comment since the present manuscript has not the aim to perform an analysis of the study designs or/and to give the level of evidence of the trials already published. The manuscript is not intended to be a meta-analysis or a Cochrane analysis. The manuscript is focusing on what is presently available to face the issue of insulin resistance in PCOS without using insulin sensitizer drugs.
Conclusions
The final conclusion has been modified according to the comment of the reviewer.
As suggested, Table 1 has been omitted from the manuscript.
Reviewer 2 Report
The manuscript expounded the endocrine profile, metabolic profile, generally pharmacological therapies and integrative treatments of polycystic ovary syndrome (PCOS). The content is rich and the demonstration basis is sufficient. The content of the manuscript has certain guiding significance for the treatment of PCOS, but it is too cumbersome and does not highlight the key points. There are some problems to be further improved as well.
1. In Endocrine profile of PCOS and Metabolic profile in PCOS sections, they had less description of insulin resistance and hyperinsulinemia, especially the Endocrine profile of PCOS section, which described the causes, mechanisms and other related contents of hyperandrogenism at large. But the title of the manuscript is “Integrative strategies to counteract insulin-resistance in PCOS patients”. Is it inappropriate?
2. In the Inositols section, the content was too complex, and it should echo the title. It is better to focus on the effect of inositol in the treatment of insulin resistance in patients with PCOS.
3. In the Alpha Lipoic Acid section, the first four paragraphs were all about the antioxidant effect of ALA, which can be briefly described.
4. In the Carnitines section, paragraphs 5-7 described the role of carnitines in other diseases, which was not associated with PCOS and could be omitted.
5. In N-acetylcysteine and L-arginine section, the logic was not very clear. In the third paragraph, NO was suddenly proposed, and the relationship between NO and NAC had not been explained clearly.
6. In Melatonin section, the elaboration of melatonin in the treatment of PCOS insulin resistance was not deep enough.
7. The first four sections of the manuscript mainly introduced the general profile and treatment of PCOS. Besides, the content of integrative treatments was also more extensive. It seemed to be inconsistent with the title of the manuscript.
8. The structure of manuscript was cumbersome. It is suggested to put similar content under one title. For example, take treatment as the first level title, and the content related to treatment as the second level title.
Author Response
Referee # 2 (changes highlighted in green)
- The comment is pertinent and the title has been modified accordingly and it has been changed in “Putative complementary compounds to counterct insulin resistance in PCOS”. In regards to the mechanisms of insulin resistance we moved most of the explanations in the sections where we discussed about the various compounds. It resulted to be the best way to explain how the integrative intervention might work. An example of this is in regards to inositols section
- The comment is correct but most of the problems of the reduced insulin sensitivity is related to an impaired function of the inositol pathway. That’s why we described carefully the relevance of this.
- The commenti s correct. We shortened the paragraphs about the antioxidant effect of ALA
- The commenti s correct. Effects of carnitines in other diseases has been omitted.
- The commenti s correct. We revised the section accordingly adding details.
- As suggested, melatonin section has been implemented.
- The title of the manuscript has been changed (see point 1)
- As suggested, titles of the sections have been modified accordingly.
Reviewer 3 Report
The authors perform a systematic review in publications in polycystic ovary syndrome (PCOS) in the etiopathology, examinations and treatments. They analyze the endocrine profile of PCOS patients and metabolic profile in PCOS women involved in insulin resistance. They go on to analyze the treatment options for PCOS patients including pharmacological therapies and integrative treatments.
This review provides the latest research progress to understand the PCOS. It also offers preferences to treat PCOS women.
1. It might be easier to understand ‘2. Endocrine profile of PCOS’ and ‘3. Metabolic profile in PCOS’ if adding a summary figure/carton for each part.
2. Line 17: ‘Therefore a great variety of integrative treatments have been proposed’. Add ‘.’ after ‘proposed’.
3. Line 372: ‘ART’ means?
4. Please remove the extra ‘weight’ in the sentence in line 898: ‘…weight reduction weight is one of the first strategies to get therapeutic results, …’
Author Response
Referee # 3 (changes highlighted in azure)
- As suggested an additional figure has been created and added (Figure 1) to give an easier access to the relationship between metabolism and the many endocrine impairments that triggers thecompensatory hyperinsulinemia
- Line 17 – As suggested we made the change
- Line 372 – The meaining of ART has been added
- Line 898 – The suggested change has been done
Round 2
Reviewer 2 Report
The manuscript entitled “Putative complementary compounds to counteract insulin-resistance in PCOS patients” expounded the endocrine profile, metabolic profile, generally pharmacological therapies and integrative treatments of polycystic ovary syndrome (PCOS). The content is rich and the demonstration basis is sufficient. The content of the manuscript has certain guiding significance for the treatment of PCOS. My previous suggestion has been adopted by the author and my concerns have been resolved. No further recommendations are currently available.